# Experimental and Numerical Research of Post-Tensioned Concrete Beams

**DOI:** 10.3390/ma16114141

**Published:** 2023-06-01

**Authors:** Anna Jancy, Adam Stolarski, Jacek Zychowicz

**Affiliations:** Faculty of Civil Engineering and Geodesy, Military University of Technology, 2 gen. Sylwestra Kaliskiego Street, 00-908 Warsaw, Poland; jacek.zychowicz@wat.edu.pl

**Keywords:** post-tensioned beams, experimental investigations, finite element method analysis, concrete damage analysis

## Abstract

The purpose of this paper is to present a new approach for the modelling of post-tensioned beams with calibration of the FE model to experimental results until the load capacity and post-critical state are reached. Two post-tensioned beams with different nonlinear tendon layouts were analysed. Material testing for concrete, reinforcing steel and prestressing steel was performed prior to the experimental testing of the beams. The Hypermesh program was used to define the geometry of the spatial arrangement of the finite elements of the beams. The Abaqus/Explicit solver was used for numerical analysis. The concrete damage plasticity model was used to describe the behaviour of concrete with different laws of elastic–plastic stress–strain evolution for compression and tension. Elastic-hardening plastic constitutive models were used to describe the behaviour of steel components. An effective approach to modelling the load was developed, supported by the use of Rayleigh mass damping in an explicit procedure. The presented model approach ensures good agreement between numerical and experimental results. The crack patterns obtained in concrete reflect the actual behaviour of structural elements at every loading stage. Random imperfections found during experimental studies on the results of numerical analyses were also discussed.

## 1. Introduction

Prestressed elements are successfully used worldwide to obtain longer spans, higher load capacities and to reduce the dead weight of structures. However, the most important factor for assessing the safety of structures is to understand the behaviour of materials and their properties, especially in accidental situations. The results of the behaviour analysis of structural elements can be obtained during experimental studies or predicted in advanced modelling using the finite element method (FEM). Even the most detailed numerical modelling cannot be treated as a relevant result. It should be calibrated with experimental tests prepared for structural element geometry, material properties and loading conditions.

Prestressed structures have been studied by many researchers, and their approaches to verifying the behaviour of structural elements are different and apply to different types of elements.

There are several published works aimed at verifying the behaviour of structural elements under the action of prestressing force on the basis of experimental studies. Noble et al. [1] studied the behaviour of post-tensioned beams in static and dynamic impact tests. Limongelli et al. [2] conducted tests on prestressed concrete beams with different tendon arrangement to verify intelligent systems for the prevention of structural hazards due to aging effects. Ng Koon and Tan Hwee [3] have analysed the influence of the span to depth ratio on the flexural behaviour of externally prestressed beams. Six et al. [4] tested the bending behaviour of three-span unbonded post-tensioned elements at a span to depth ratio of 45. Rawi et al. [5] have compared the behaviour of prestressed slabs with traditionally reinforced concrete slabs under low-velocity impacts. Kim et al. [6] conducted tests on full-size specimens of PSC bridge girders to obtain friction characteristics of post-tensioning tendons. Szydłowski and Łabuzek [7] analysed shrinkage, creep and prestressing losses in post-tensioned concrete beams as a result of using a lightweight aggregate obtained by sintering fly ash.

Some of the work was aimed at confirming that the new theoretical approaches lead to a good agreement between the results of analytical and experimental studies. Fu et al. [8] developed a method for assessing the stiffness of existing cracked post-tensioned concrete beams with unbonded tendons. Kim and Lee [9] developed a method of nonlinear analysis for continuous post-tensioned elements, and then compared the obtained results with the results of experimental tests.

Several authors have calibrated the FEM results using experimental investigations or empirical formulations. There are various modelling techniques to model prestressed concrete elements, either 2D or 3D. In the publications, reinforcing steel bars are often modelled as truss elements (T3D2) embedded in the concrete material. Tendons are usually defined as truss elements, 2-node beam elements (B31) or solid elements (C3D8). Approaches to modelling stress–strain relationship for structural materials also differ. Some researchers do not include tension stiffening for concrete in the analysis and some of them do not consider the plastic hardening of steel.

Kang et al. [10] studied the behaviour of post-tensioned concrete elements with bonded and unbonded tendons. Three-dimensional structural models were developed to compare the results with experiments. The concrete material, tendon ducts and steel plates were modelled with solid elements. Truss finite elements were used to model the reinforcing steel. The interaction between the tendon and concrete was modelled with contact definition between surfaces. The concrete damage plasticity (CDP) model was used, but without specifying the damage parameters. Hafezolghorani et al. [11] developed a simplified concrete damage plasticity model and tested it in the FE model of a prestressed concrete beam with a straight tendon modelled with solid elements. Concrete material was also modelled as solid, but steel rebars were truss elements in 3D space. A nonlinear softening path was introduced in the concrete model. Brenkus et al. [12] developed FE models of a post-tensioned I-girder bridge construction with various types of fillers. Pre-tensioning and post-tensioning tendons were modelled as truss elements, while the interaction of tendons with the concrete was modelled with a virtual tendon. A nonlinear approach for tension stiffening was adopted. Ellobody and Bailey [13] developed a nonlinear 3D FE model to study the effects of changing the type of aggregate, load ratio and boundary conditions on the behaviour of a post-tensioned concrete slab loaded at elevated temperatures. The obtained results were verified by experimental trials. FEM models consisted of concrete, straight unbonded tendons and anchorages modelled as solid elements. The CDP model was used with concrete linear elastic behaviour limited to 0.33$\;f_{c}$ and tensile failure stress of 0.1$\;f_{c}$. Lee et al. [14] investigated the effect of external post-tensioning of a steel bar system using the CDP model to verify the concrete damage behaviour and then compared the numerical results with experimental tests. Authors obtained good agreement between numerical and experimental results in terms of the crack patterns, displacement and strength of the post-tensioned beams. Strengthening members and concrete were modelled as shell elements. Truss elements were used as internal reinforcement, and 2-node linear beam elements were used as external bars. Embedding conditions for the steel rebar were used, and surface contact between concrete and saddle pin was defined. Tensile failure stress was assumed at 0.1$\;f_{c}$, with an ultimate tensile strain of 10$\;f_{t}/E_{c}$. The linear behaviour of concrete in the softening phase was assumed. Huang et al. [15] verified two modelling approaches to simulate unbonded conditions of post-tensioned slabs. The first consisted of contact techniques reflecting the real physical state, and the second of a multiple-spring system, which proved to be more flexible in modelling due to the convergence of the solution. Bonded steel reinforcing bars were modelled with truss elements embedded in the concrete material, and post-tensioned tendons were modelled with 2-node beam elements (B31). The CDP model was used for concrete, but without damage modelling during the unloading process. The developed FE results were compared with experimental results. A good agreement of moment to deflection responses and damage patterns between the numerical spring system model and the experimental results of post-tensioned slab specimens were obtained. Tran et al. [16] developed a 3D FEM model of precast dry-joined segmented concrete girders internally prestressed with unbonded FRP tendons to verify flexural behaviour of the beams. The concrete material, tendons and anchorages were modelled using solid elements, and the reinforcing steel bars using truss elements. The authors defined load in the models by forcing the displacement. The compressive stress of 0.4$\;f_{c}$ was assumed as the elastic limit of concrete. Pham and Hong [17] studied the evolution of strains in post-tensioned beams, with bonded or unbonded curved tendon systems, using the CDP model for concrete. Numerical stability was achieved by introducing an artificial damping factor and viscosity parameter. In the FEM beam models, both tendons and concrete have been modelled as solid elements. Longitudinal steel reinforcing bars were also modelled with solid elements, while the stirrups were created as truss members. The limit of the linear elastic behaviour of concrete at the level of 0.4$\;f_{c}$ and nonlinear softening in tension were assumed. Wang et al. [18] analysed the ductility and resistance of prestressed concrete beams reinforced with an internal H-steel section, using FEM models and load–deformation curves. The multi-span system of tendons and stirrups was modelled using truss finite elements. The H-steel section was modelled with shell finite elements (S4R) and concrete material was defined as a solid finite element. The contact surfaces of H-steel section and the tendons embedded in the concrete was modelled as the bonding condition. It was assumed that the elastic limit of concrete is reached at 0.5$\;f_{c}$. Cornejo et al. [19] developed a 3D FEM model of post-tensioned concrete elements to study the reduction of prestressing stress in tendons as a result of steel relaxation. Structural elements were modelled as a composite material whose behaviour was described by the serial-parallel rule of mixtures (S/P RoM). Stress relaxation was simulated using the generalized Maxwell visco-elastic model. Various models have been developed where tendons are modelled as truss finite elements embedded in the concrete material.

FEM models are also used to verify the local behaviour of different parts of a prestressed structure in the event of their actual failure in order to investigate the cause of the loss of load capacity.

Chrościelewski et al. [20] investigated the cause of concrete cracking during tensioning in the areas of the anchorage head. A numerical model of a single section of the bridge structure was developed. The concrete material was modelled as a solid finite element, anchorages—as shell finite elements and steel reinforcing bars—as truss elements. For concrete material, the CDP model with a linear softening path was used. Setiawan et al. [21] analysed the behaviour of the anchorage zone of the existing bridge section during the tensioning process. The FEM model has also been developed with concrete and anchorage plates modelled as solid finite elements, and with steel reinforcing bars modelled as truss finite elements. An embedded interaction element was used between the concrete and the tendon. Huang and Kang [22] developed FEM models to verify the sliding behaviour in post-tensioned members. The authors considered unbonded, partially bonded and fully bonded conditions. The nonlinear FEM code was developed in a MATLAB environment. Post-tensioned tendons were modelled as nonlinear truss finite elements embedded in the concrete, which was modelled as a two-node Euler–Bernoulli beam finite element.

The following observations emerge from the bibliography review. Combined experimental studies and finite element method modelling for post-tensioned structures are carried out by relatively small group of researchers. Tests and analyses of post-tensioned beams are usually limited to a straight tendon route. Comparative analysis of the behaviour of post-tensioned beams with different curvilinear tendon systems in combined experimental studies with numerical modelling were not the subject of research. The approach to the creation of finite element meshes is similar for all material components. However, concrete is most often modelled as a solid element, the anchorages of the tendons are modelled in a simplified way as flat elements and the reinforcing bars are modelled as truss elements in a 2D plane or in 3D space. Due to the approach to modelling adopted in other publications, the local loss of concrete stiffness caused by the specific location of reinforcing bars, especially stirrups, was not taken into account in the modelling of crack propagation in concrete. Damage analysis is usually performed with a concrete damage plasticity (CDP) material model implemented in Abaqus. However, no numerical analyses take into account the damage of concrete that is already under compression from the beginning of the plastic hardening phase. In addition, no detailed description of the development of concrete damage in particular deformation phases of post-tensioned beams is provided, which is important for understanding the behaviour of beams under the influence of external loads.

The aim of this article is to present a new approach to the modelling of post-tensioned beams with a detailed calibration of the FEM model to the experimental results until the load capacity and post-critical state are reached. Two post-tensioned beams with curvilinear systems of tendons with different eccentricities were experimentally tested. As part of the experimental studies, strength tests of concrete, prestressing steel and reinforcing steel bars were also performed. The finite element method for beam models was developed using Abaqus/Explicit approach. Hypermesh was used to define the finite element mesh. A spatial model of the entire beam was developed without applying symmetry conditions. Hexahedral finite elements were used to model all construction materials. The finite elements of the curvilinear tendons and all reinforcing bars were surrounded by the finite elements of the concrete. The anchorages were modelled as multi-segment steel blocks. Appropriate contact conditions of finite elements for concrete with lateral surfaces of finite elements for all steel structural elements were also modelled. The concrete damage plasticity model was used to describe the behaviour of concrete. A four-phase model of stress–strain evolution for compression and a three-phase model for tension was proposed. These models included an extended range of damage description from the onset of plastic hardening in compression and in the range of constant residual stress both in compression and tension. The method proposed in the papers of Cichorski and Stolarski [23,24] and Stolarski et al. [25] was used to estimate the value of stresses and strains of concrete at the phase boundaries of the stresses and strains evolution model, both in compression and tension. Elastic—hardening plastic constitutive material models, were used to describe the behaviour of steel tendons, rebars and anchorages. The static, load—displacement analysis of a post-tensioned beam was obtained using an explicit solution procedure with load application modelling and selection of the Rayleigh mass-damping coefficient. The comparative analysis carried out showed a good agreement between the numerical results and the experimental results. A detailed analysis of the stress state in the concrete, reinforcing steel, tendon and anchorages was presented. Particular attention was paid to explaining the development of concrete damage in each phase of the behaviour of post-tensioned beams. The possibility of modelling the influence of unintended, random material imperfections found during experimental studies on the results of numerical analyses was also discussed.

## 2. Experimental Research

### 2.1. Post-Tensioned Concrete Beams

Two post-tensioned beams were designed. Their cross-section size was 150 mm × 220 mm from C40/50 designed concrete class. Both beams were 3.2 m long. Main prestressing reinforcement was assumed as a single-strand 7-wire tendon with a 15.2 mm diameter and a total cross-section of 139 mm^2^, the characteristic tensile strength declared by the manufacturer was $f_{pk}$ = 1860 MPa with a minimum ultimate capacity of $P_{ult}\;$= 259 kN. An unbonded prestressing system was used in both beams (low friction between the tendon and the concrete, without usage of injection grout). Two different tendon profiles were designed in each beam. The tendon in the first beam (beam No. 1, Figure 1a) was designed with a 300 mm straight part along its length at the beam faces to avoid any additional bending moments at the anchorage zone and above the supports. Then, the tendon profile was parabolically changed to achieve 25 mm eccentricity measured from the centreline of the beam’s cross-section to the tendon’s centre. The second beam (beam No. 2, Figure 1b) had a similar approach in terms of the principles of the tendon layout (straight part at the beam faces); however, it was parabolically changed to achieve 55 mm eccentricity in the mid-length of the beam. For details of the analytical design of the beams, see Jancy et al. [26].

Mono-strand TTM 1E15 anchorages were used in both experimental beams, Figure 2.

Additionally, beams were reinforced with steel reinforcing bars. Two 10 mm diameter bars were located in both the upper and lower parts of the section. The cross-sectional area of these reinforcing bars was designed as minimum reinforcement area required to control cracking. Both post-tensioned beams were designed in a such a way that the ultimate load capacity was determined by the strength of the concrete compression zone. Shear capacity was increased by stirrups of 6 mm diameter at the 60 mm spacing in the support zone and then with 105 mm spacing within the central part of the beam. At the anchorage zones, spiral reinforcement and additional stirrups were provided according to the European Technical Approval of the manufacturer anchorages, Tension Technology Martin S.r.l. [27]. Reinforcement, tendons and anchorages are presented in the Figure 3.

### 2.2. Loading and Testing Program

During concreting of the beams, 5 samples of 150 mm × 150 mm × 150 mm concrete cubes were taken for compressive strength test according to PN-EN 206 + A1:2016-12 [28].

On the 28th day, 2 concrete samples were tested, the formwork of the beams was removed, and the tendon was prestressed to the first level of prestressing force of 158 kN.

On the 45th day, i.e., on the day of beam testing, the concrete strength of 3 samples was checked and the tendon was prestressed to the second level of prestressing force of 206.1 kN.

### 2.3. Structural Materials Testing

#### 2.3.1. Concrete

The machine controls of I-st accuracy class were used to conduct the testing. The results of compressive strength tests $f_{ci,cub}$ and the mean compressive strength $f_{cm,\;cub}$ for the concrete cube samples, are presented in the Table 1.

According to the compliance criteria for the compressive strength test specified in PN-EN 206 + A1:2016-12 Annex B, [28]: “the test result should be the average of the results obtained on at least two test samples made of one sample of the mixture and tested at the same age”. Therefore, the compliance criteria for the designed concrete class C50/60 have been met.

#### 2.3.2. Prestressing Steel

A tensile strength test was performed for three samples taken from the prestressing strand. This strand was used to transfer prestressing force into concrete in the beams. Experimental results for tensile strength $f_{p0.1}$ registered at 0.1% permanent strain and tensile ultimate strength limit $f_{pt}$, modulus of elasticity $E_{p}$ and elongation at failure $\varepsilon_{pt}$ were obtained from tests. The results with their mean values are presented in the Table 2.

Figure 4 presents the stress—strain relationship obtained from the experimental strength verification of the prestressing bars.

### 2.4. Reinforcement Steel Bars

Reinforcing steel grade B500SP, class C with increased ductility was used. The results of the reinforcement steel bars’ tensile strength tests are presented in Table 3.

### 2.5. Stand and Method for Strength Testing of the Beams

Ultimate capacity of the beams was verified using the ZD40 strength machine of the I-st accuracy class with a maximum load application up to 400 kN. For the purposes of the tests, additional steel beams for both supporting and loading zones were introduced to allow for testing of 3200 mm long beams. Based on the following, a four-point bending scheme was adopted with two loading and two support points, as shown in Figure 5.

Loading was applied with a constant value of maximum 0.8 kN/s. Results were recorded with Phantom MIRO M310 camera with a recording speed of 3260 fps at a maximum resolution of 1280 × 800 pixels and high-resolution digital camera Nikon D5200 with a maximum photo resolution of 6000 × 4000 pixels. Photos were taken every one second.

### 2.6. Results of the Experimental Research of Strength Testing of the Beams

Both beams were loaded until the ultimate strength of the concrete was achieved. For each beam test, damage progression was similar. Firstly, uniformly distributed perpendicular cracks appeared at the largest bending moment area (Figure 6 and Figure 7), then a local damage of the compression zone was achieved (Figure 8 and Figure 9) and finally after the ultimate load was achieved, the steel reinforcing bars buckled, which caused further concrete degradation within both compression and tension zones (Figure 10). The obtained behaviour of the beams reflects the design assumptions regarding the ultimate capacity depending on the compressive strength of the concrete.

The highest concentration of degradation within the compression zone appeared between loading points for beam No. 1. For beam No. 2, it was localized under the left loading point. It has been noted that the loading beam had rotated during the loading force application process, which could be a reason for the damage concentration at the left loading point instead of at the middle part of the beam.

Figure 11 presents obtained results of the loading force to displacement relationship for both beams. Ultimate force value for beam No. 1 was $Q_{u1}$ = 80.90 kN, the value occurred at $u_{1}$ = 36.60 mm of vertical displacement. For beam No. 2, ultimate capacity was equal to $Q_{u2}$ = 94.80 kN with $u_{2}$ = 53.11 mm displacement achieved.

## 3. Numerical Models of the Post-Tensioned Concrete Beams

### 3.1. Numerical Model Description

Due to multiple curvatures caused by assumed tendon layout, a proper model meshing has a crucial impact on the obtained results. In addition, a post-tensioned beam has anchorages on both ends to maintain prestressing force in the beam, which has further implications on the FE modelling approach. The developed models consist of a tendon in a nonlinear route layout, two anchorages on both ends, steel reinforcing bars (both longitudinal bars and stirrups) and concrete material. The finite elements of all the beams’ component parts were created as C3D8R spatial elements in the Hypermesh and then mesh geometry was exported to Abaqus/Explicit to perform the computational analysis, shown in Figure 12.

The model of beam No. 1 has 121,951 nodes and the model of beam No. 2 has 114,121 nodes. The finite elements of tendon, steel reinforcing bars and concrete material stiffnesses are not duplicated, so the model stiffness is not artificially increased. Each component part of each beam contains its own finite elements with the corresponding contact parameters defined between each of them, as shown in Figure 13.

Tendon with anchorages and steel reinforcement were modelled as a separate FE-s as shown in the Figure 14.

### 3.2. FE Mesh Details

The mesh size of the beam model was dependent on the geometry and curvature of the tendon in different areas of the beam, the location of the anchorages and the location of the steel reinforcement. Generally, a 20 mm × 20 mm mesh grid was provided along longitudinal direction, but mesh density was reduced to a 10 mm × 10 mm grid between the loading blocks. At the stirrups’ location, a 5 mm × 5 mm mesh was provided. Additionally, mesh elements located next to the stirrups in the zone between loading blocks were also meshed with a denser mesh of 5 mm × 5 mm. Two areas of denser mesh were provided for the elements adjacent to the tendon (above and below the tendon). Those were meshed with 5 mm × 5 mm elements, as shown in Figure 15a.

Beam No. 2, due to the higher inclination angle of the tendon, had smaller element sizes for the mesh elements adjacent to the tendon within areas with the largest tendon eccentricity. Here, the smallest element size was 4 mm × 4 mm, show in Figure 15b.

### 3.3. Materials Models Definition

#### 3.3.1. Prestressing Steel

The defined values of the stresses and strains for prestressing bars are calibrated with experimental results presented in Section 2.3.2.

Constitutive model of prestressing steel is defined by elastic and nonlinear plastic material properties. The four-linear stress–strain diagram was properly adjusted to the experimental curve shown in Figure 16.

The following characteristic points of the stress—strain diagram were assumed:Elastic strength limit;Lower yield limit equal to the tensile strength at 0.1% permanent plastic strain;Upper yield limit;Ultimate tensile strength.

Corresponding stress and strain values at each characteristic point are presented in the Table 4.

#### 3.3.2. Steel Components

For the other steel components present in the FE model, i.e., reinforcing bars and anchorages, a catalogue of values of strength and deformation parameters were adopted based on the manufacturer’s information, Tension Technology Martin S.r.l. [27], with the exception of strength values for reinforcing steel, for which these parameters have been tested, see Table 3.

Based on the modulus of elasticity and yield strength values, the elastic strain values were calculated. Steel properties for anchorages have the largest ultimate strain values, which indicate the highest ductile level of all presented steel types. The strength and deformation parameters of steel components are presented in the Table 5.

A constitutive model of steel components is defined by elastic and linear plastic hardening material properties. Based on the adopted strength and deformation parameters, bilinear, elastic–plastic with linear hardening and stress–strain relationships, are presented in Figure 17.

#### 3.3.3. Concrete

The concrete damage plasticity model was used for numerical analysis. There are several publications by Jankowiak et al. [29], Sumer and Aktas [30], Voyiadjis and Taqieddin [31], Yu et al. [32], Bernkus et al. [12] and Szczecina and Winnicki [33], in which the basic model approach is explained, therefore in this article the authors will focus on the parameters of the concrete material model, established only in considered analysis.

At first, the basic parameters of the CDP model were assumed, such as dilation angle ψ, plastic flow potential eccentricity ε, ratio of initial compressive strength in an equal biaxial stress state to initial uniaxial compressive strength $\sigma_{b0}/\sigma_{c0}$, ratio of the second stress invariant on the tensile meridian $K_{c}$ and viscosity parameter μ. Table 6 presents the assumed material model parameters.

From the above-mentioned parameters, the dilation angle ψ and the eccentricity of the plastic flow potential *ε* have the greatest impact on the obtained results. An important stage of the analysis is the appropriate calibration of the model parameters in relation to the experimental results. The dilation angle was analysed in the range of 10°≤ψ≤ 56.3°. Smaller values of the dilation angle are more appropriate for structural elements in high shear or at brittle failure, while higher values better reflect damage to the ductile material. Similar studies were carried out on the influence of the eccentricity of the plastic flow potential in the range of 0.1≤ε≤1.0 and the influence of the ratio of the second stress invariant on the tensile meridian in the range of $0.5\leq K_{c}\leq 1.0$ on the numerical results.

Then, the stress–strain relationships for compression and tension together with the damage variables, were determined. The basis for the description of these relationships is the determination of the mean values of concrete’s compressive and tensile strength as well as the deformation modulus. For this purpose, the authors assume that all concrete parameters relevant to the analysis depend on the mean compressive strength $f_{cm,cub}$ for the concrete cube samples, which was obtained from the tests and is presented in Table 1. The procedure of the parameters of actual concrete class determination is based on both the partial experimental results as well as on the parameters of two adjacent classes of concrete defined by the standard [34]. In this sense, the actual concrete class has an interpretation as a theoretically possible strength class.

Therefore, if the actual concrete class:(1)$$Ca=Cf_{ck,cyl,a}/f_{ck,cub,a}$$
is an intermediate class between two consecutive standard classes of concrete:(2)$$C0=Cf_{ck,cyl,0}/f_{ck,cub,0}\;\mathrm{and}\;C1=Cf_{ck,cyl,1}/f_{ck,cub,1}$$
this assuming that:(3)$$f_{ck,cub,a}=f_{cm,cub}$$
the following value can be determined by linear interpolation:(4)$$f_{ck,cyl,a}=f_{ck,cyl,0}+\frac{f_{ck,cub,a}-f_{ck,cub,0}}{f_{ck,cub,1}-f_{ck,cub,0}}\left(f_{ck,cyl,1}-f_{ck,cyl,0}\right)$$
which is the basis for determining the mean values of the concrete compressive and tensile strengths, and the deformation modulus, [34]:(5)$$f_{cm}=f_{ck,cyl,a}+8$$
(6)$$f_{ctm}=\left\{\begin{array}{ccc} 0.3{\left(f_{ck,cyl,a}\right)}^{2/3} & if & C\leq C50/60\\ 2.21\;\ln(1+0.1f_{cm}) & if & C>C50/60 \end{array}\right.$$
(7)$$E_{cm}=22{\left(f_{cm}/10\right)}^{0.3}$$

Taking into account that the mean compressive strength for the concrete cube samples is $f_{cm,cub}=61.4\;\mathrm{MPa}$, Table 1, the two adjacent standard classes of concrete are C0=C50/60 and C1=C55/67. Hence, the actual concrete class is Ca=C51.0/61.4, Equation (1). As a result, the mean compressive strength is $f_{cm}=59.0\;\mathrm{MPa}$, Equation (5), the mean tensile strength is $f_{ctm}=4.095\;\mathrm{MPa}$, Equation (6)_2_, and the mean deformation modulus is $E_{cm}=37.47\;\mathrm{GPa}$, Equation (7).

Concrete material is very sensitive to creep effects. It is often found that FE models obtain much stiffer behaviour than experimental beams; however, even if the experiment happens just a few days after pouring the concrete, creep and shrinkage effects occur. This is one of the reasons why the authors have found it important to include creep effects in the FE analysis. This approach is based on the assumption that the total deformation including creep is calculated by using an effective deformation modulus for concrete according to the formula, EN 1992-1-1 [34]:(8)$$E_{cm,eff}=\frac{E_{cm}}{1+\varphi \left(t,t_{0}\right)}$$
where: $\varphi \left(t,t_{0}\right)$ is the creep coefficient relevant for the load and time interval.

The creep coefficient is calculated using the method provided in EN 1992-1-1 [34], in the Annex B—creep and shrinkage strain.

Considering that the beams were supported by a formwork until compressive strength was reached, and were prestressed to the first prestressing force level in $t_{0}=28\;\mathrm{day}$, beam tests were performed in t=45 day after pouring the concrete, and the relative humidity was RH=50%, the authors have obtained a creep coefficient equal to $\varphi \left(t,t_{0}\right)=0.78$, and an effective deformation modulus for concrete equal to $E_{cm,eff}=21.05\;\mathrm{GPa}$. By including creep coefficient in the numerical analysis, deformation modulus is reduced by approximately 44%, which has a significant impact on the beam’s behaviour.

The most crucial factor for the proper definition of the constitutive relationship in the concrete damage plasticity material model is the adoption of the appropriate law of stress evolution in the strain function. For this purpose, separate laws of evolution were adopted for compression and tension, as shown in Figure 18.

For compression, an elastic–plastic material model with non-linear hardening, linear softening and constant residual stress was assumed. For tension, the elastic material model with linear softening and constant residual stress was adopted.

This model is described by the following stress–strain relationships:(9)$$\sigma_{c}=\left\{\begin{array}{ccc} -f_{ctr} & \;\mathrm{if}\; & \varepsilon_{c}\leq -\varepsilon_{ctr}\\ -\left[f_{ctr}-E_{cts}\left(\varepsilon_{c}+\varepsilon_{ctr}\right)\right] & \;\mathrm{if}\; & -\varepsilon_{ctr}\leq \varepsilon_{c}\leq -\varepsilon_{ctf}\\ E_{c}\varepsilon_{c} & \;\mathrm{if}\; & -\varepsilon_{ctf}\leq \varepsilon_{c}\leq \varepsilon_{ce}\\ f_{ce}+\left(f_{c}-f_{ce}\right)\frac{k\eta -\eta^{2}}{1+(k-2)\eta } & \;\mathrm{if}\; & \varepsilon_{ce}\leq \varepsilon_{c}\leq \varepsilon_{cf}\\ f_{c}+E_{cs}\left(\varepsilon_{c}-\varepsilon_{cf}\right) & \;\mathrm{if}\; & \varepsilon_{cf}\leq \varepsilon_{c}\leq \varepsilon_{cr}\\ f_{cr} & \;\mathrm{if}\; & \varepsilon_{c}\geq \varepsilon_{cr} \end{array}\right.$$
where: $f_{c}$ is the compressive strength, $f_{ce}=\beta_{ce}f_{c}$ is the compressive elastic limit, $\beta_{ce}$ is the elastic range coefficient in compression, $f_{cr}=\;\beta_{cr}f_{c}$ is the residual compressive strength, $\beta_{cr}$ is the residual strength range coefficient in compression, $\varepsilon_{ce}=\frac{f_{ce}}{\;E_{c}}$ is the elastic strain limit in compression, $E_{c}$ is the elastic deformation modulus, $\varepsilon_{cf}$ is the strain limit in the range of plastic hardening in compression, $\eta =\frac{\varepsilon_{c}-\varepsilon_{ce}}{\varepsilon_{cf}-\varepsilon_{ce}}$ is the relative strain in the range of plastic hardening in compression, $k=E_{c}\frac{\varepsilon_{cf}-\varepsilon_{ce}}{f_{c}-f_{ce}}$ is the proportionality coefficient of the deformation modules in the range of plastic hardening in compression, $E_{cs}=-\frac{f_{c}-f_{cr}}{\varepsilon_{cr}-\varepsilon_{cf}}$ is the deformation modulus in the range of plastic softening in compression, $\varepsilon_{cr}$ is the strain limit in the range of plastic softening in compression, $f_{ct}$ is the tensile strength, $\varepsilon_{ctf}=\frac{f_{ct}}{\;E_{c}}$ is the elastic strain limit in tension, $f_{ctr}=\;\beta_{ctr}f_{ct}$ is the residual tensile strength, $\beta_{ctr}$ is the residual strength range coefficient in tension, $E_{cts}=-\frac{f_{ct}-f_{ctr}}{\varepsilon_{ctr}-\varepsilon_{ctf}}$ is the deformation modulus in the range of plastic softening in tension and $\varepsilon_{ctr}$ is the strain limit in the range of plastic softening in tension.

A full description of the adopted law of stress evolution in the strain function for concrete modelling is possible with the following parameters, Table 7.

More detailed information on the modelling of concrete behaviour is presented, among others, in the works of Cichorski and Stolarski [23,24] and Stolarski et al. [25].

The last step of the CDP material model definition is a proper assignment of the damage parameters for both compressive and tensile zones. It was assumed that the damage parameter is proportional to the maximum stress value that material is capable to withstand to the actual residual stress which occurs.

In accordance with this assumption, the damage model is described by the following damage–stress relationships:(10)$$d=\left\{\begin{array}{ccc} d_{t\max} & \;\mathrm{if}\; & \varepsilon_{c}\leq -\varepsilon_{ctr}\\ d_{t\max}\frac{-\varepsilon_{c}-\varepsilon_{ctf}}{\varepsilon_{ctr}-\epsilon_{ctf}} & \;\mathrm{if}\; & -\varepsilon_{ctr}\leq \varepsilon_{c}\leq -\varepsilon_{ctf}\\ 0 & \;\mathrm{if}\; & -\varepsilon_{ctf}\leq \varepsilon_{c}\leq \varepsilon_{ce}\\ d_{c\mathrm{mmax}}\frac{\varepsilon_{c}-\varepsilon_{ce}}{\varepsilon_{cf}-\varepsilon_{ce}} & \;\mathrm{if}\; & \varepsilon_{ce}\leq \varepsilon_{c}\leq \varepsilon_{cf}\\ d_{cS\min}+\left(d_{c\mathrm{mmax}}-d_{c\min}\right)\frac{\varepsilon_{cf}-\varepsilon_{c}}{\varepsilon_{cf}-\varepsilon_{cr}} & \;\mathrm{if}\; & \varepsilon_{cf}\leq \varepsilon_{c}\leq \varepsilon_{cr}\\ d_{c\max} & \;\mathrm{if}\; & \varepsilon_{c}\geq \varepsilon_{cr} \end{array}\right.$$
where: $d_{tmax}$ and $d_{cmax}$ are the maximum values of the damage parameter in tension and in compression, respectively, $d_{cHmax}$ is the maximum value of damage parameter in the range of plastic hardening in compression, $d_{cSmin}$
and $d_{cSmax}$ are the minimum and the maximum values of damage parameter in the range of plastic softening in compression, respectively.

A full description of the adopted damage model is possible with the following limit values of damage parameters:(11)$$d_{cHmax}=d_{cSmin}=0.05,\;d_{tmax}=d_{cmax}=d_{cSmax}=0.9$$

Figure 19 shows the implemented stress–strain relationship with the corresponding damage variables. There are characteristic horizontal lines of the stress—strain diagram describing the residual stresses. The advantage of such modelling is to ensure the greatest ductility of the concrete material in interaction with steel reinforcement in reinforced concrete structure to enable extended damage simulation within the FE model.

#### 3.3.4. True Stress—Strain Relationships

All experimental, code and analytical stress and strain relationships discussed above are considered nominal. For the purposes of numerical analysis, they have been transformed into true stress–strain relationships, based on the equations:(12)$$\varepsilon_{true}=ln\left(1+\varepsilon_{nom}\right)$$
(13)$$\sigma_{true}=\sigma_{nom}\left(1+\varepsilon_{nom}\right)$$
where: $\varepsilon_{nom}$—nominal strain and
$\sigma_{nom}$—nominal stress.

### 3.4. Contact Definition

Anchorages, stirrups and longitudinal rebars are permanently connected with concrete material, without any slippery action. The surface-to-surface contact was used as a contact discretization method, consisting of the equality of displacements in the nodes of both contacting surfaces. The same assumption was made for contact between the outer surface of the tendon and the inner surface of the hole in the anchorage.

Due to the assumed unbonded post-tensioned system in the experimental research, similar behaviour needed to be reflected in the numerical model. In this case, the interaction between the outer surface of the tendon and the inner concrete surface at the contact points is defined as the kinematic contact method with finite sliding. Contact interaction properties were defined in tangential and perpendicular directions to the tendon. Within a tangential direction, the friction formulation was used with a friction coefficient of μ=0.06, taken according to the manufacturer specification, Tension Technology Martin S.rl. [27].

In turn, in the perpendicular direction, the contact pressure can be defined alternatively for two cases. Namely, if any contact occurs, the contact pressure is transferred between the defined surfaces, or if surfaces do not touch or separate after contact, the contact pressure drops to zero. In the numerical analysis it was assumed that the tendon is tightly embedded with concrete, and that it is not possible to separate the two materials in a perpendicular direction.

### 3.5. Explicit Solving Procedure

Static analysis of post-tensioned beam will be obtained using an explicit solving procedure based on limit formulation of the dynamic equilibrium equation of motion.

#### 3.5.1. Dynamic Equilibrium Equation for Static Problems Analysis

An explicit procedure was used for solving the equation of motion under the assumption of the lumped mass matrix of diagonal structure, see Wriggers [35]. Then, the accelerations at the beginning of the current time were calculated as:(14)$$\ddot{w}\left(t\right)=M^{-1}\left[P\left(t\right)-C\dot{w}\left(t\right)-Kw\left(t\right)\right]$$
where: M, C and K are the nodal mass, damping and stiffness matrices, $\ddot{w}\left(t\right)$, $\dot{w}\left(t\right)$ and $w\left(t\right)$ are the nodal accelerations, velocities and displacements vectors and $P\left(t\right)$ is the vector of the external applied forces.

The Rayleigh model of damping was adopted, in which the damping matrix is a linear combination of the mass and stiffness matrices:(15)C=αM+βK
where α and β are assumed parameters.

The central-difference scheme is used for explicit time integration the equations of motion. Within this scheme the velocities and the displacements are determined at the end of the next time increment t+∆t knowing all the kinematic conditions from the previous increments:(16)$$\dot{w}\left(t+∆t\right)=\dot{w}\left(t-∆t\right)+2∆t\ddot{w}\left(t\right)$$
(17)$$w\left(t+∆t\right)=w\left(t\right)+∆t\frac{\dot{w}\left(t+∆t\right)+\dot{w}\left(t\right)}{2}$$

The initialization the central-difference scheme necessitates the introduction of the initial conditions at the time $t_{0}=0$:(18)$$\ddot{w}\left(t_{0}\right)={\ddot{w}}_{0},\;\dot{w}\left(t_{0}\right)={\dot{w}}_{0},\;\dot{w}\left(t_{0}-∆t\right)={\dot{w}}_{0}-∆t{\ddot{w}}_{0}$$
where: the relation (18)_3_ follows on the first-order accurate Taylor-series expansion for the velocities.

Since the explicit method for integration equations of motion is conditionally stable, it is necessary to introduce the time step limitation in relation to the critical time step:(19)$$\Delta t\leq \Delta t_{crit}=\frac{\Delta L_{min}^{e}}{c_{d,max}}$$
where: $\Delta L_{min}^{e}$ is the smallest characteristic length of the element in the FE-discretization, $c_{d,\;max}=\sqrt{\frac{E}{\rho }}$ is the fastest dilatation wave velocity in the elastic material, E is Young’s modulus and ρ is the mass density.

#### 3.5.2. Damping Parameters

According to the Rayleigh damping model, structural damping is defined as proportional to both mass and stiffness of structure.

In the numerical analysis, only Rayleigh mass damping was used, changing the α parameter in the range $\alpha =\left[100;2000\right]$. The omission of the structural stiffness damping (*β* = 0) is justified by the fact that non-linear material models are used, which describe the significant part of the structure stiffness reduction.

Comparative analysis of the numerical force—displacement results with the results of experimental tests of the beams will allow for the selection of the optimal mass-damping parameter.

#### 3.5.3. Dilatation Wave Velocity and Critical Time Step

Decisive for calculation of the dilatation wave velocity for the entire post-tensioned concrete beam model is the material defined for anchorages with parameters E=200 GPa and $\rho =7.85E-09\;t/{\mathrm{mm}}^{3}$, which yield the fastest dilatation wave velocity $c_{d,\;max}=5047544.65\;\mathrm{mm}/s$.

The smallest characteristic length of the element in the FE-discretization for both beam models is equal to $∆L_{min}^{e}=2.15\;\mathrm{mm}$ (the element is located between the loading points, next to the tendon). Based on the Equation (19) the critical time step value is equal to $∆t_{crit}=4.26E-07\;s$.

#### 3.5.4. Prestressing Force Value

The post-tensioned beam has initial loading caused by prestressing force in the tendon. Prestressing force was applied in the beam numerical model as a result of initial stress in the tendon.

Since the beams were ultimately prestressed on the day of the experimental trial, it was assumed that only initial prestressing losses occurred without any long-term effects, e.g., steel relaxation.

Total prestressing force was reduced by initial losses and then, as the value equal to $P_{m0}=185.76\;\mathrm{kN}$, it was converted to an initial stress with value of
$\sigma_{p,m0}=1336.4\;\mathrm{MPa}$, and applied in the beam model.

#### 3.5.5. The Method of Applying the External Load

Since the purpose of the analysis is to obtain the behaviour of a static model, not a dynamic one, the method of applying the external load has a significant impact on the behaviour of the numerical model. Too high a load application velocity may cause an unexpected model response.

It has been found that the most effective way to control the loading process is to use the vectors of displacement velocity propagation, defined in every node on the top surface of selected beam nodes as presented in the Figure 20.

Typically, the recommended static load velocity value should be less than 1% of the fastest dilatation wave velocity, as proposed by the manual Abaqus [36].

However, this proposed range turned out to be too large for post-tensioned beams. At this velocity of loading application, large oscillations occurred in the model. It should be noted that a typical post-tensioned beam already has an initial prestressing force applied, which results in a greater dynamic response than typical reinforced concrete beams with no initial forces applied.

In order to avoid model oscillations and the so-called local jetting effect, the blocks displacement velocity applied cannot exceed 0.025% of the fastest dilatation wave velocity identified in the model of beam. For this proportion of the fastest dilatation wave velocity, the value of displacement velocity was applied as $\upsilon_{load}=1261.89\;\mathrm{mm}/s$.

Of equal importance to the value of displacement velocity, in order to stop the oscillating motion and to reflect the slow and smooth loading application, is the determination of the load amplitude and the total analysis time. For the considered static analysis, the method of smooth steps loading application was found to be suitable.

Total analysis time depends on the expected total displacement value and loading velocity:(20)$$t_{tot}=\frac{u_{max}}{\upsilon_{load}}$$

Since the maximum load capacity of the beams was reached experimentally with the displacement
$u_{max}=u_{2}=53.11\;\mathrm{mm}$ for beam No. 2 (Figure 11), if the load in the numerical models continues until the assumed displacement is reached, e.g., 70 mm, then the total analysis time is $t_{tot}=0.055\;s$, and the number of analysis steps is equal to $n=\frac{t_{tot}}{∆t_{crit}}=129108$.

The relationship of the load amplitude to the analysis time was proposed using the smooth steps method in the following form, see manual Abaqus [36]:(21)$$\upsilon =\upsilon_{load}\left\{\begin{array}{ccc} \tau^{3}\left(10-15\tau +6\tau^{2}\right) & if & 0\leq t\leq t_{c}\\ 1 & if & t_{c}\leq t\leq t_{tot} \end{array}\right.$$
where: $\tau =\frac{t}{t_{c}}$ is the relative time parameter, $t_{c}$ is the time to reach a maximum load velocity value.

The applied relationship of the load velocity amplitude to the analysis time is shown in the Figure 21.

Comparative analysis of the numerical results of the force—displacement relationship with the results of experimental tests of the beams will allow for the selection of the optimal value of the time
$t_{c}$ in the range of $t_{c}=\left(0.1;0.9\right)t_{tot}$.

## 4. Results of the Numerical Research

### 4.1. Comparative Analysis of the Numerical and Experimental Results for the Force—Displacement Relationship

The obtained numerical results of the force–displacement relationship are largely dependent on the selected mass-damping parameter. Several calculation tests were performed according to the mass-damping parameter range presented in Section 3.5.2. In the presented numerical analysis, the most appropriate mass-damping parameter has been found to be at the level of ∝ = 500, for which the results were closest to the experimental research results. Both beams reached the maximum displacement between the load points, and the lower middle node (see Figure 5) was used to compare the course of the displacement during the loading process. Figure 22 shows the dependence of the load force on the displacement obtained from the numerical analysis of both beams. The load capacity of beam No. 1 was reached with the loading force $Q_{u1}\;$= 80.64 kN, and this value occurred with the vertical displacement
$u_{1\;}$= 36.50 mm. The load capacity of beam No. 2 was $Q_{u2}\;$= 96.50 kN with displacement $u_{2\;}$= 36.44 mm.

Basically, both diagrams reflect the experimentally found behaviour of the beams during the loading process. Particularly good agreement between the numerical results and the experimental results was found for beam No. 1. However, in the numerical solution of beam No. 2, the stiffness of the beam to reach the load capacity was greater than in the experimental test. Namely, a good agreement of the load capacity was obtained with a much smaller value of displacement.

The reason for the lower agreement of the numerical and experimental results in beam No. 2 can be seen in the probable occurrence of unintentional, random material imperfections in the experimental beam. This fact is confirmed by the almost identical stiffness of both beams observed in the experimental diagrams until the load value of about 65 kN is reached. This means that the potentially stronger beam No. 2 showed no greater stiffness than beam No. 1.

### 4.2. Numerical Analysis of Beams Behaviour

The highest values of the maximum principal stresses are obtained below the velocity application nodes. Negative maximum principal stresses reflect compression stresses, and positive values represent tensile stresses (in MPa units). The highest stress values are obtained between loading points, but they do not exceed the strength of the material. Just below the load application areas there are local stress concentrations due to local load application. A similar stress distribution is obtained for both beams after reaching their load capacity (Figure 23).

The first tensile cracks occurred in the concrete below the load application points, in the place of stirrups in both beams. Further damage propagation consists of the appearance of successive cracks between the stirrups, and then in diagonally distributed regions. Figure 24 shows the state of cracking under a load of 45 kN, which reflects the damage immediately after the elastic—linear phase. Beam No. 1 had more cracks than beam No. 2 at a similar load.

Differences in tensile damage propagation were observed for both beams. Beam No. 1 for its hardening phase reached larger diagonal damage areas located on the outer sides of the load points than beam No. 2 (Figure 25).

The extent of these zones increased in the post-critical phase of softening. Moreover, beam No. 1 had larger areas of tensile damage next to the load points than beam No. 2 (Figure 26).

Since the damage was described from the beginning of plastic hardening in compression, the maximum range of compression damage under load capacity was obtained in the numerical results shown in the Figure 27. The damage areas in beam No. 1 are much larger than those in beam No. 2. These areas are especially related to the zones outside the loading forces, where the highest values of shear forces occur.

In the post-critical softening phase, beam No. 1 had larger areas of compression damage located outside the loading forces than beam No. 2. In turn, beam No. 2 had larger zones of compression damage between loading forces than beam No. 1, (Figure 28).

The yield strength was reached only in the lower longitudinal rebars, in the regions below the loading forces (Figure 29). The yield limit was not reached in any stirrup.

Figure 30 shows the maximum principal stresses in the tendons for half of the longitudinal section. Compared to the initial stress value of 1336 MPa, the stresses in the tendons increased beyond the limit of elastic strength (point No. 1 in Figure 16), but do not exceed the lower yield strength equal to the tensile strength at 0.1% of permanent plastic strain (point No. 2 in the Figure 16), when both beams reached their load capacity. The greatest stresses occur in the straight part of tendon located between loading forces.

Figure 31 shows the stresses in one of the anchorages for each beam. The anchorage in beam No. 1 has lower tensile stresses at the contact surface with concrete than the anchorage in beam No. 2.

### 4.3. Discussion on Imperfections Analysis

#### 4.3.1. Location of Damage Initiation

The damage progression obtained from numerical analysis was fully symmetrical, since no imperfections were introduced in the models. In fact, reinforced concrete structures experience many imperfections caused by, e.g., uneven distribution of aggregate, blockage of aggregate between reinforcement bars, rotation of stirrups during concrete pouring or local reduction of concrete cover. Such imperfections may cause an initial cross-sectional reduction; this is the starting point for further damage progression during loading.

Asymmetrical damage progression occurred in both experimental beams. The initial cracks in beam No. 1 appeared in the middle between loading points (Figure 8), while in beam No. 2 the damage was located below the left loading application area (Figure 9). The location of the damage in beam No. 1 did not affect the total values of displacement. In beam No. 2 there were differences in the plastic hardening phase—more ductile behaviour than in the numerical model, as shown in the Figure 22.

There was no significant difference in the load capacity of both beams in the experimental results in the elastic phase and in the plastic hardening phase (up to 65 kN loading force), despite the introduction of the different eccentricities of the tendon system. Actually, beam No. 2 should be stiffer than beam No. 1. There may have been additional imperfections related to the execution of the beams, the precise preparation of the stand or the execution of the experiment, which could have caused the unexpected behaviour of beam No. 2. The numerical results show the expected differences in load capacity from the initial load phases.

Imperfection modelling can be initiated by introducing an artificial crack in the place obtained during experimental tests. A similar approach is also presented by other authors, e.g., Jankowiak and Łodygowski [29] and Szczecina and Winnicki [33]. For beam No. 1, the greatest concentration of compressive force was in the middle between load points. Therefore, a similar crack initiation location was selected for the numerical model of the beam, as shown in Figure 32.

The initiated crack caused the greatest concentration of stresses between the load points (Figure 33) and resulted in the highest values of tensile and compressive damage parameters in the middle of beam No. 1. The damage parameters at the load capacity are shown in Figure 34 and Figure 35.

The maximum stresses in the steel reinforcement have also changed. Now, instead of only stress concentrations below the load points, the maximum stress values occurred below the crack initiation point as shown in Figure 36.

#### 4.3.2. Local Concentrations of Stresses

There are a few specific places where local stress values exceed the material resistances determined in the applied constitutive model of concrete. These places are located near the load application points and at the greatest contact forces of the steel reinforcement with the concrete in both numerical models. In experimental studies, the loading force was applied locally by a steel support beam. Therefore, authors decided to model a similar approach in the numerical analysis. Due to the nature of the load application, stress values above the concrete compressive strength appeared in the overload zone at the achieved beam load capacity, as shown for the location of the right loading point in beam No. 1 in the Figure 37. A very similar effect occurred symmetrically near the left loading point in this beam.

In the post-critical phase of the beam deformation, a similar effect was observed at the contact place between the steel reinforcement and the concrete (Figure 38). At this place, the upper longitudinal steel reinforcing bars began to bend.

These specific overload areas must be the focus of further detailed contact analysis. The behaviour of concrete in these areas is influenced, for example, by the size and shape of the FE mesh, the area (or volume) of the contact zone between two materials with extremely different properties and determination of the properties of concrete with increased strength parameters in high compression contact zones.

## 5. Conclusions

This paper presents the results of the numerical analysis of post-tensioned beams based on the results of its own experimental research.

On the basis of the obtained results, the following conclusions can be drawn, relevant for conducting experimental research, modelling of post-tensioned beams using the FEM and calibrating the numerical model to the experimental results:
Precise preparation of experimental models of post-tensioned concrete beams and careful execution of experimental tests allowed the avoidance of unintentional model imperfections and the obtainment of appropriate results;The use of a separate finite element for modelling each material component of each beam: concrete, prestressing steel, reinforcing steel and anchorages, with appropriate modelling of contact interactions, results in a damage simulation qualitatively similar to the results of experimental tests. Initial cracks in the concrete occur at the location of stirrups and then develop into larger damage with new cracks between the stirrups;The convergence of the numerical solution is closely related to the precise modelling of the finite element geometry system. Elements in the finite element mesh should be made as square as possible to avoid numerical effects, i.e., shear locking and volume locking. The proposed meshing of the model allowed the avoidance of the numerical hourglass effect. The use of the reduced integration method for 3D hexahedral finite elements was sufficient to obtain a convergent solution, which is more efficient than other methods in terms of computational time. The applied meshing did not cause differences in the results between the incompatible modes and the reduced mode of the integration methods;The explicit procedure is an effective method for modelling post-tensioned concrete beams with different curvilinear tendon systems under static loading. With the ap-propriate calibration of the numerical model in relation to a detailed analysis of the experimental damage mechanism, the presented approach allows for the modelling of various geometries of post-tensioned beams. The proposed loading procedure in explicit solution method, based on an application of the loading velocity equal to 0.025% of the fastest dilatation wave velocity, gave a convergent solution for analysed post-tensioned beams without model oscillations and the local jetting effect. The beam models were calibrated to experimental results with the Rayleigh mass-damping parameter α=500;An appropriate simulation of concrete damage was obtained on the basis of the description of concrete material softening with a model of concrete damage plasticity. The inclusion of residual stresses in the concrete model ensured an increase in the ductility of the concrete material in interaction with the steel reinforcement to enable an extended damage simulation in the FEM beam models. Taking into account the effective modulus of the elasticity of concrete with the use of the real creep coefficient ensured a reduction in the stiffness in the numerical beam models by about 44%;A very good agreement between the numerical results and the experimental results for beam No. 1 was obtained. The slightly lower agreement between the numerical and experimental results for beam No. 2, especially in terms of displacements, was due to random imperfections found in the experimental results. The experimentally observed nature of concrete damage mechanisms and the numerical simulations of stress and damage states of both considered beams confirmed the design assumption about the ultimate load capacity that was determined by the strength of the concrete compression zone.

## Figures and Tables

**Figure 1 materials-16-04141-f001:**
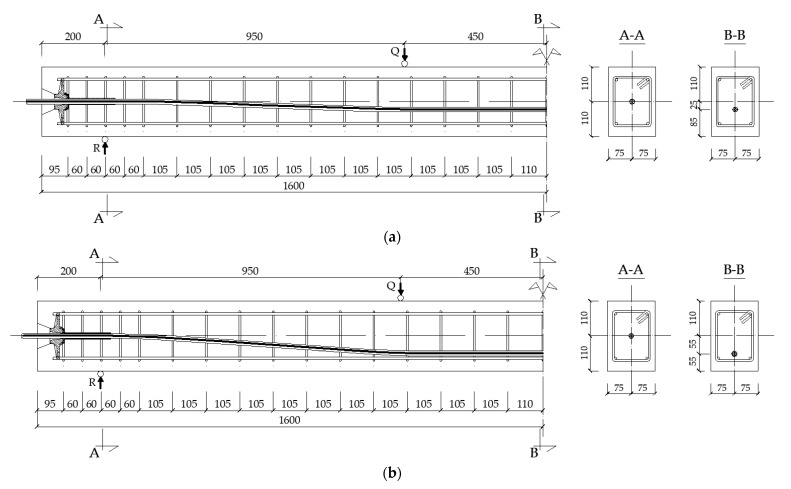
Geometry of the beams and tendon layout: (**a**) beam No. 1 with 25 mm tendon eccentricity and (**b**) beam No. 2 with 55 mm tendon eccentricity.

**Figure 2 materials-16-04141-f002:**
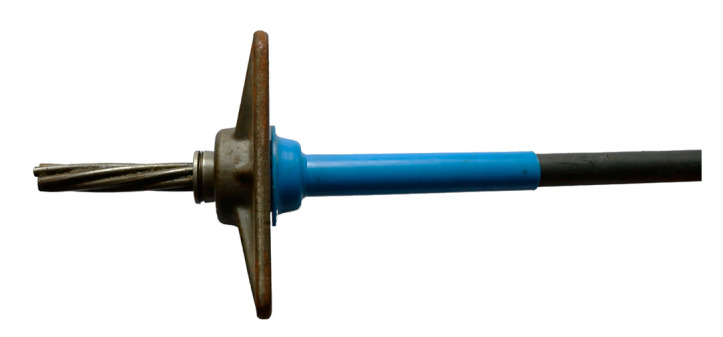
TTM 1E15 tendon anchorage.

**Figure 3 materials-16-04141-f003:**
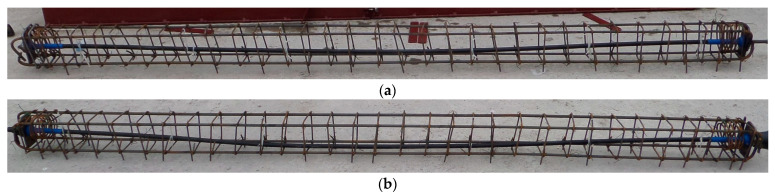
Tendon with anchorages and steel reinforcement: (**a**) beam No. 1 and (**b**) beam No. 2.

**Figure 4 materials-16-04141-f004:**
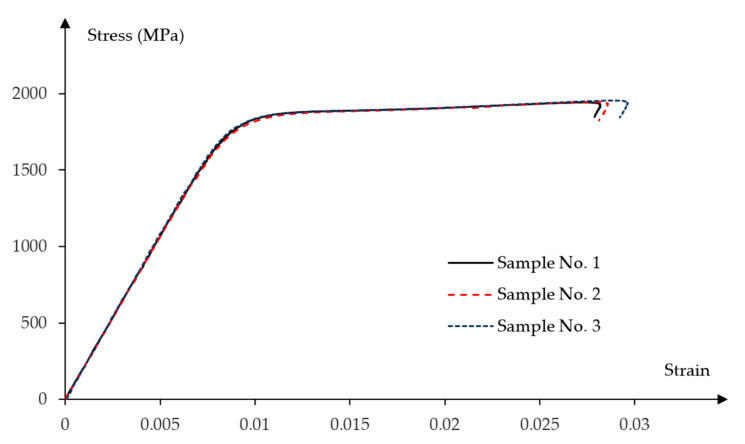
Stress–strain experimental relationships for the prestressing steel.

**Figure 5 materials-16-04141-f005:**
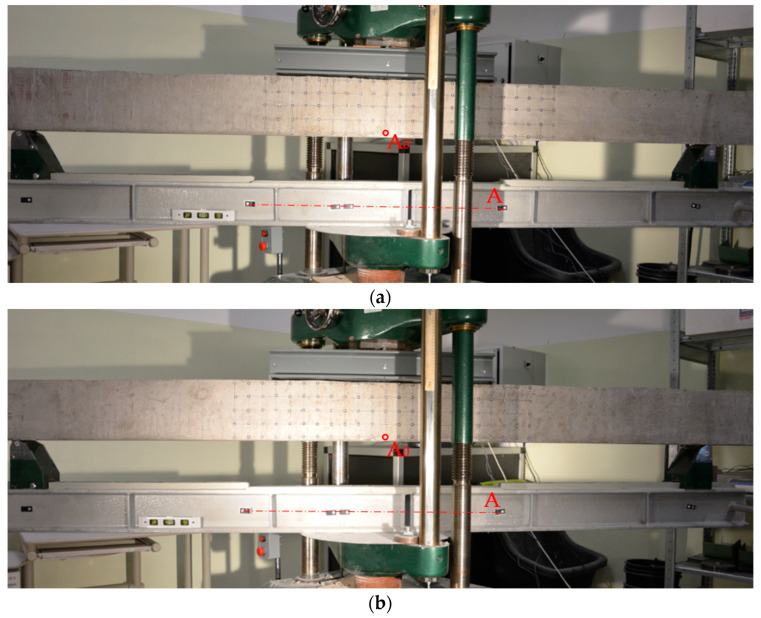
Beams on the load capacity test stand: (**a**) beam No. 1 (NOTE: the beam was placed on the test stand symmetrically in relation to the centre point (A_0_) of beam A; the grid of measurement points on the lateral plane of the beam was asymmetrical) and (**b**) beam No. 2.

**Figure 6 materials-16-04141-f006:**
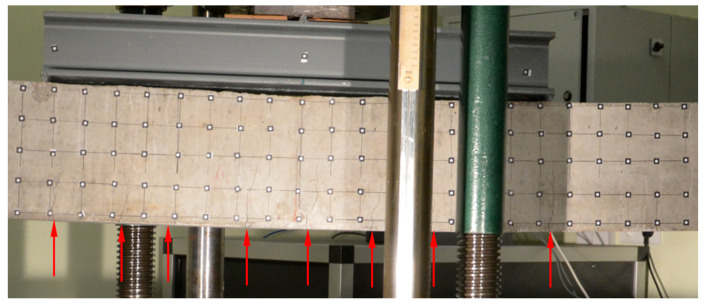
Initial perpendicular cracks propagation in beam No. 1.

**Figure 7 materials-16-04141-f007:**
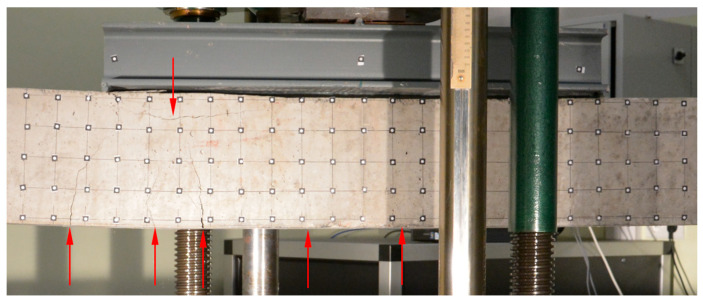
Initial perpendicular cracks propagation and horizontal crack below left loading point in beam No. 2.

**Figure 8 materials-16-04141-f008:**
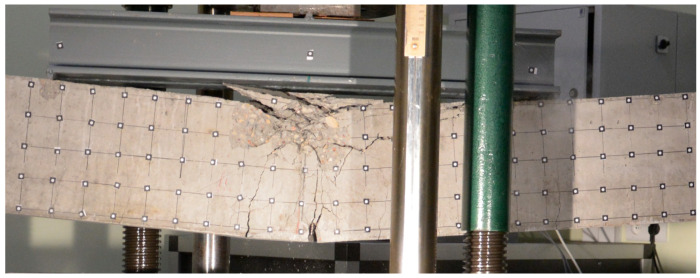
Compressive zone failure in beam No. 1.

**Figure 9 materials-16-04141-f009:**
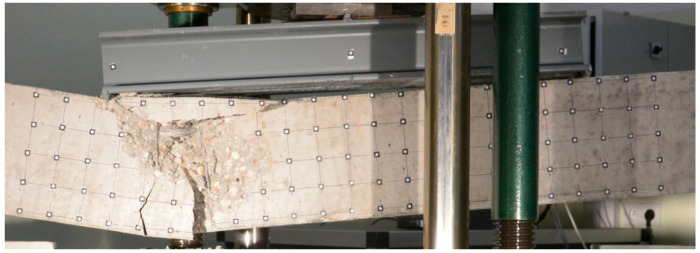
Compressive zone failure in beam No. 2.

**Figure 10 materials-16-04141-f010:**
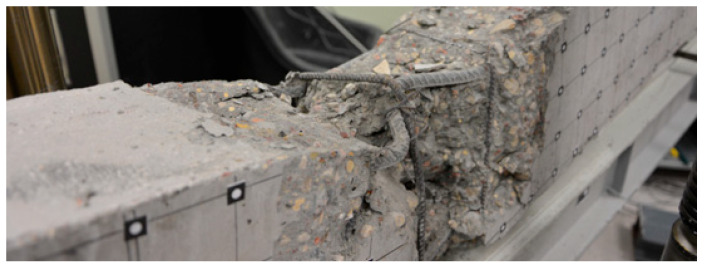
Local buckling of the steel reinforcing bars in beam No. 1.

**Figure 11 materials-16-04141-f011:**
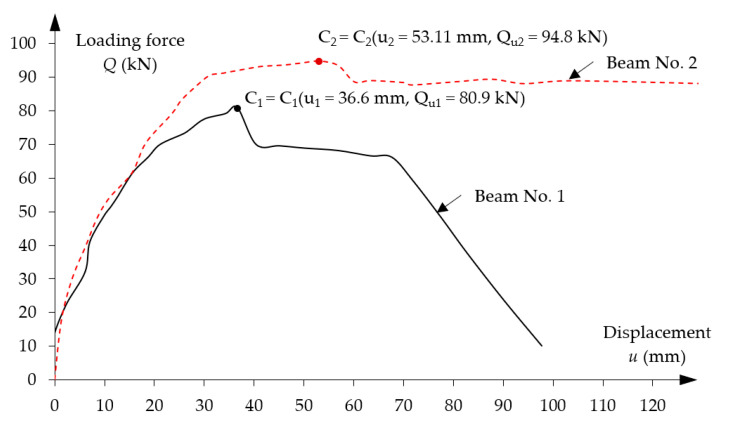
Experimental curves for loading force vs. displacement of the measuring point A.

**Figure 12 materials-16-04141-f012:**
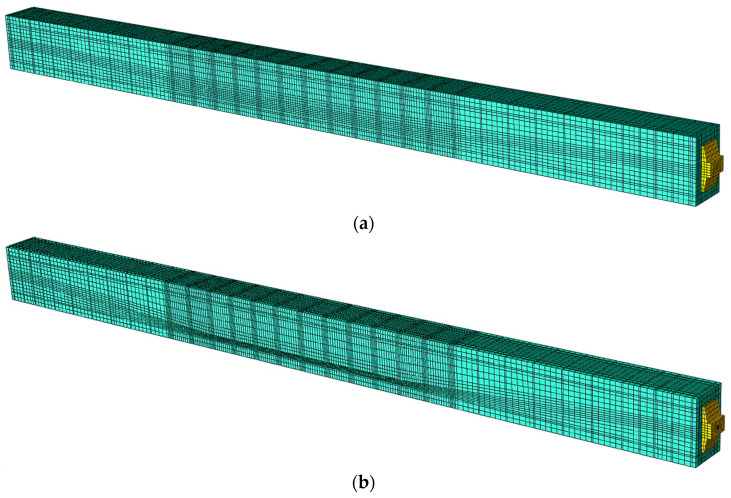
FE spatial computational model: (**a**) beam No. 1 and (**b**) beam No. 2.

**Figure 13 materials-16-04141-f013:**
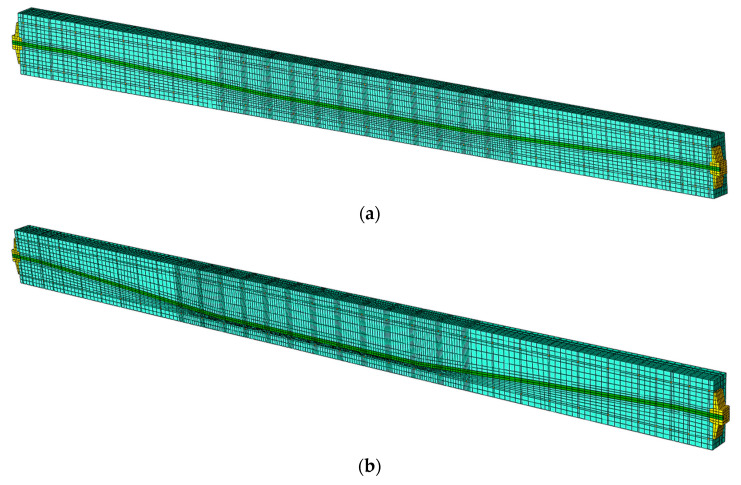
Central longitudinal cross section through FE model: (**a**) beam No. 1 and (**b**) beam No. 2.

**Figure 14 materials-16-04141-f014:**
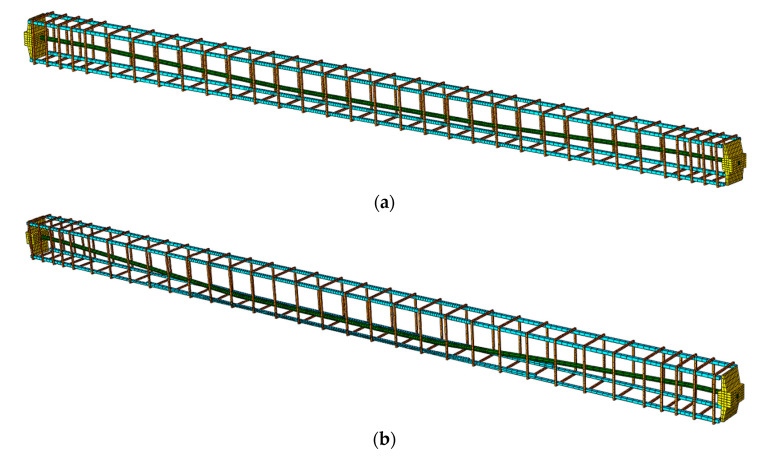
FE mesh details of steel reinforcement and tendon with anchorages: (**a**) beam No. 1 and (**b**) beam No. 2.

**Figure 15 materials-16-04141-f015:**
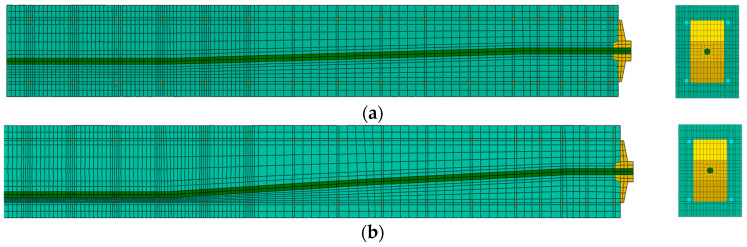
The FE mesh quality between the anchorage and the centre of the beam, longitudinal section through the middle of the beam and front view: (**a**) beam No. 1 and (**b**) beam No. 2.

**Figure 16 materials-16-04141-f016:**
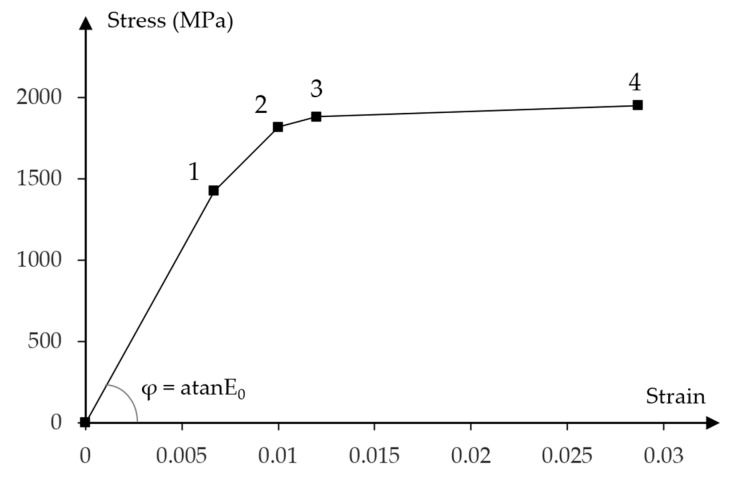
Stress to strain relationship for prestressing steel.

**Figure 17 materials-16-04141-f017:**
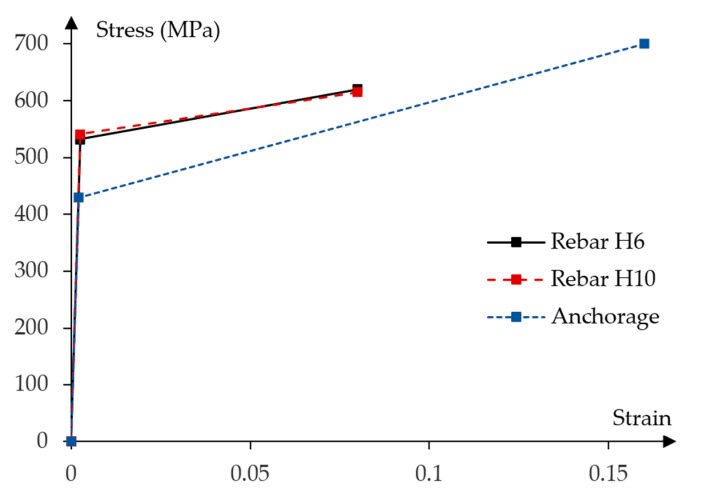
Stress to strain relationship for steel components.

**Figure 18 materials-16-04141-f018:**
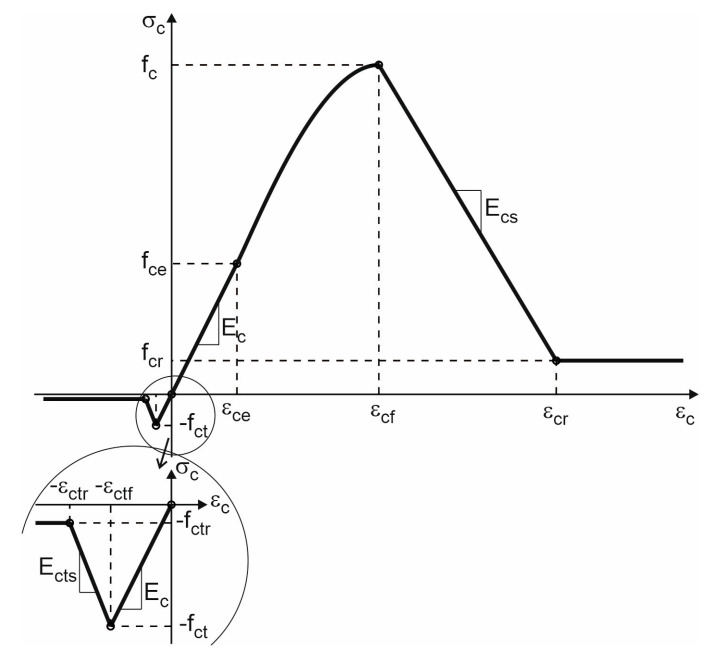
Description of compression and tension behaviour of concrete in CDP model.

**Figure 19 materials-16-04141-f019:**
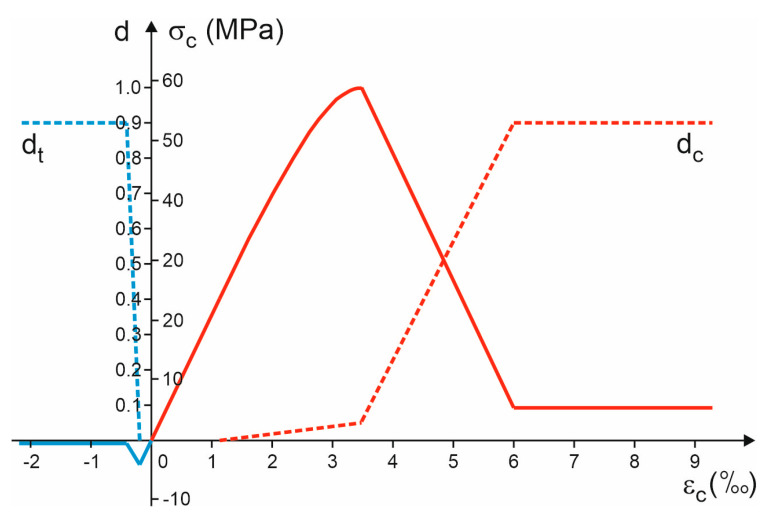
Stress to strain relationship and damage variables for concrete.

**Figure 20 materials-16-04141-f020:**
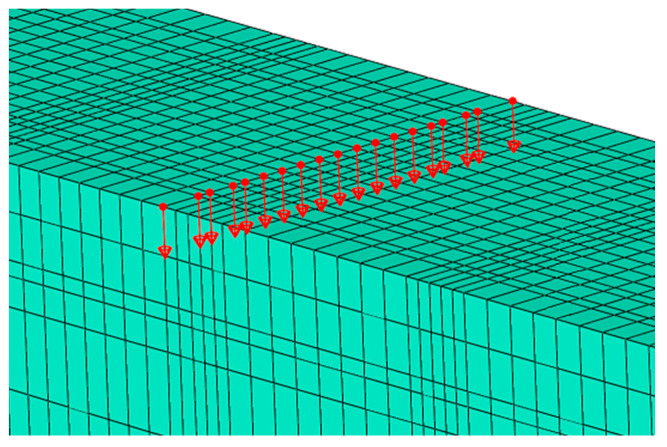
Displacement velocity propagation vectors for the loading process.

**Figure 21 materials-16-04141-f021:**
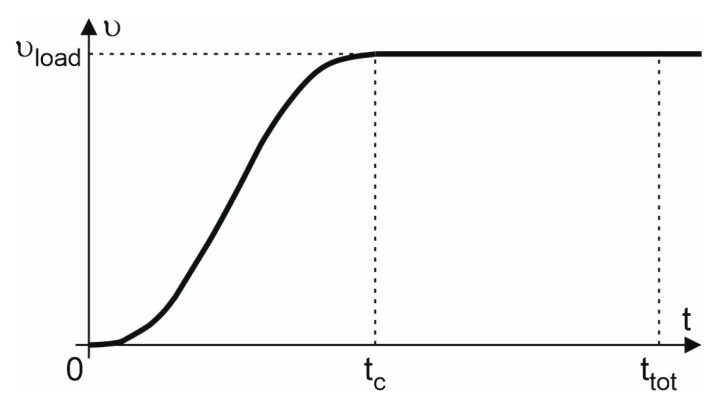
Load velocity amplitude to analysis time relationship.

**Figure 22 materials-16-04141-f022:**
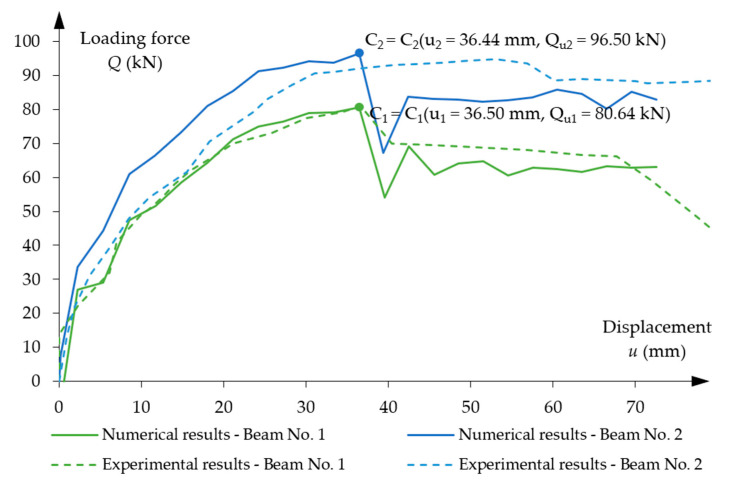
Loading force vs. displacement diagrams for beam No. 1 and beam No. 2.

**Figure 23 materials-16-04141-f023:**
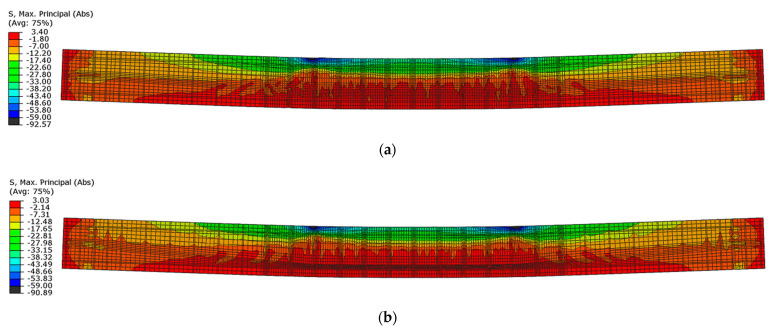
Maximum principal stresses in concrete at load capacity: (**a**) beam No. 1 and (**b**) beam No. 2.

**Figure 24 materials-16-04141-f024:**
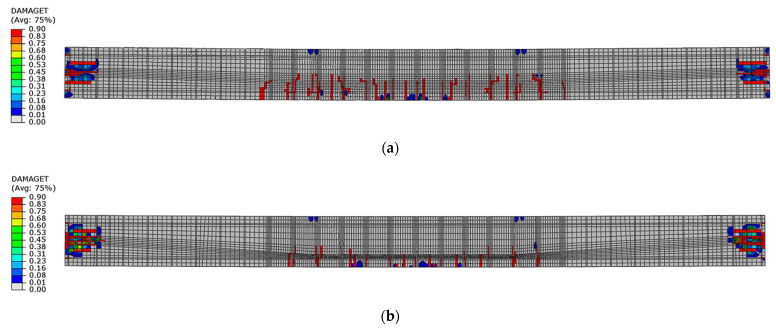
Tensile damage distribution in concrete under a load of 45 kN: (**a**) beam No. 1 and (**b**) beam No. 2.

**Figure 25 materials-16-04141-f025:**
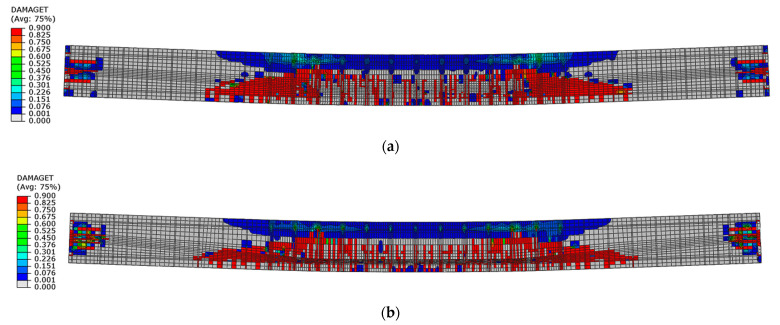
Tensile damage distribution in concrete at load capacity: (**a**) beam No. 1 and (**b**) beam No. 2.

**Figure 26 materials-16-04141-f026:**
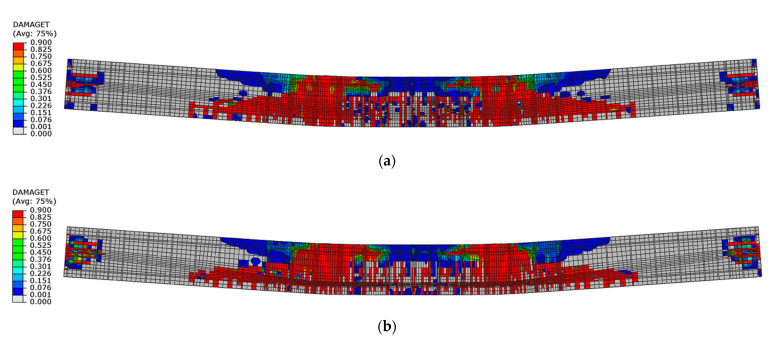
Tensile damage distribution in concrete in softening phase at displacement 70 mm: (**a**) beam No. 1 and (**b**) beam No. 2.

**Figure 27 materials-16-04141-f027:**
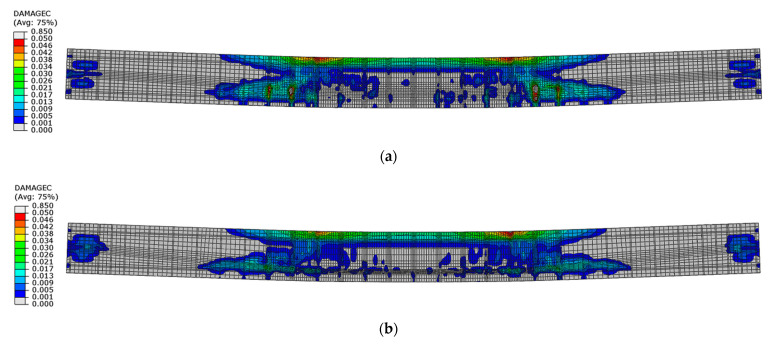
Compression damage in concrete at load capacity: (**a**) beam No. 1 and (**b**) beam No. 2.

**Figure 28 materials-16-04141-f028:**
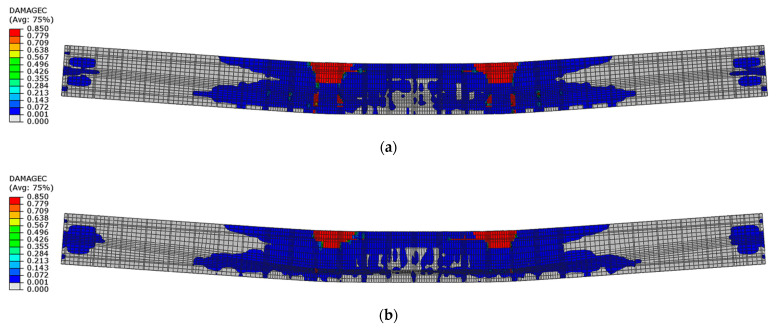
Compression damage in concrete in softening phase at displacement 70 mm: (**a**) beam No. 1 and (**b**) beam No. 2.

**Figure 29 materials-16-04141-f029:**
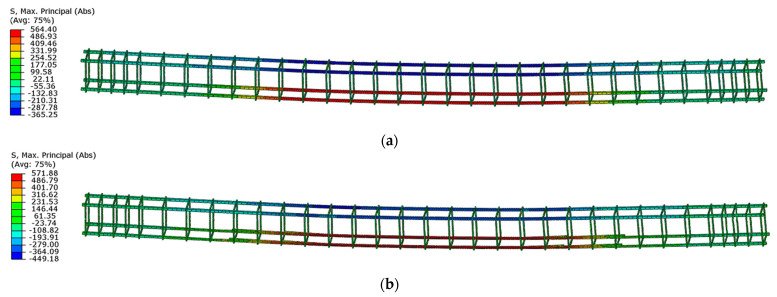
Maximum principal stresses in the steel rebars at load capacity: (**a**) beam No. 1 and (**b**) beam No. 2.

**Figure 30 materials-16-04141-f030:**
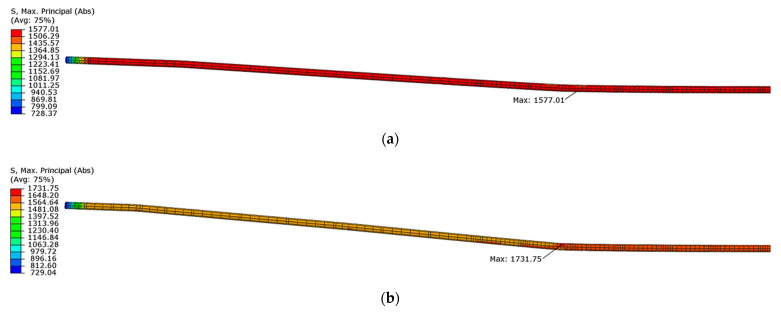
Maximum principal stresses in tendons at load capacity: (**a**) beam No. 1 and (**b**) beam No. 2.

**Figure 31 materials-16-04141-f031:**
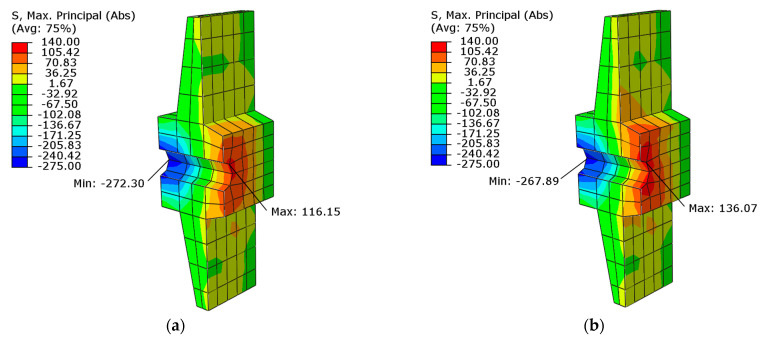
Maximum principal stresses in the anchorage in mid-section at load capacity: (**a**) beam No. 1 and (**b**) beam No. 2.

**Figure 32 materials-16-04141-f032:**
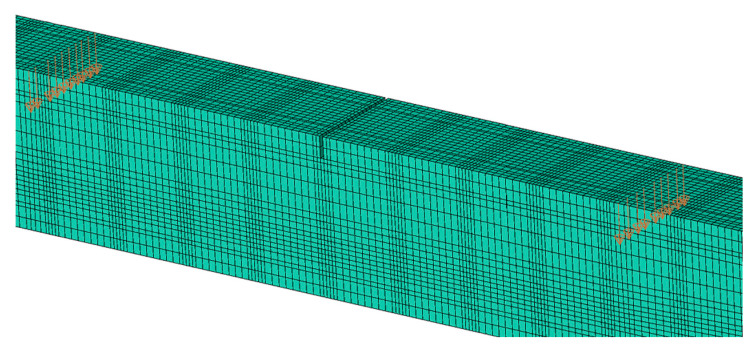
Location of crack initiation site.

**Figure 33 materials-16-04141-f033:**
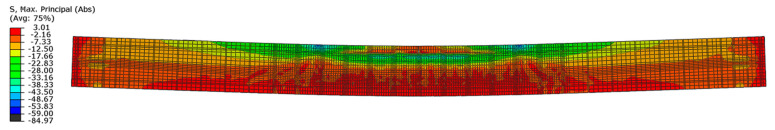
Maximum principal stresses in concrete at load capacity with crack initiated in beam No. 1.

**Figure 34 materials-16-04141-f034:**
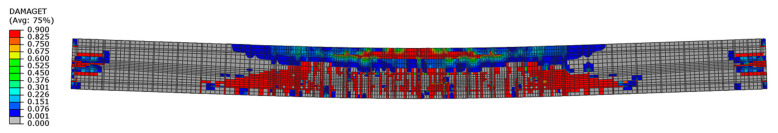
Tensile damage in concrete at load capacity with crack initiated in beam No. 1.

**Figure 35 materials-16-04141-f035:**
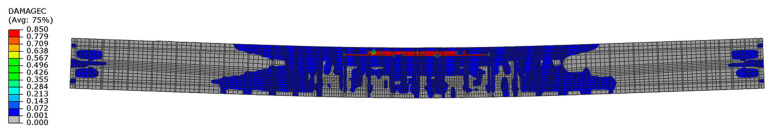
Compressive damage in concrete at load capacity with crack initiated in beam No. 1.

**Figure 36 materials-16-04141-f036:**
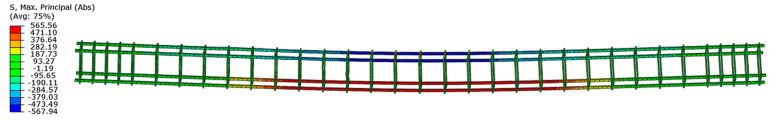
Maximum principal stresses in the steel reinforcement at load capacity with a crack initiated in beam No. 1.

**Figure 37 materials-16-04141-f037:**
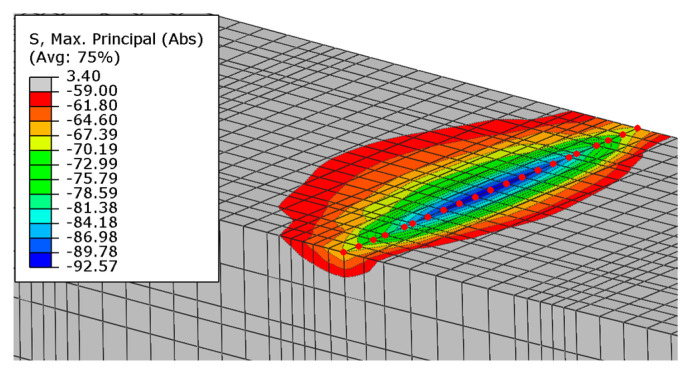
Stress concentrations next to the right loading point at the load capacity in beam No. 1.

**Figure 38 materials-16-04141-f038:**
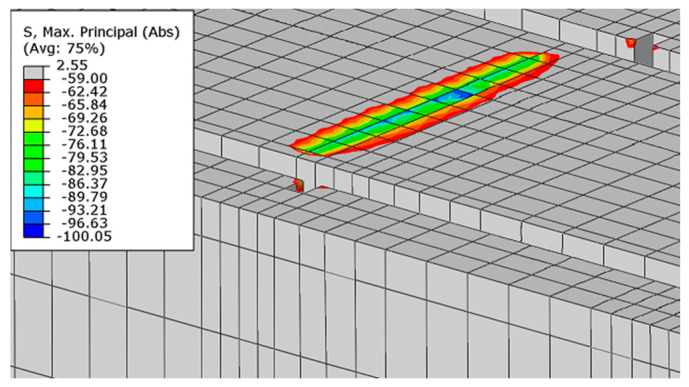
Stress concentrations next to the right loading point between longitudinal rebars in the post-critical phase of beam deformation in beam No. 1.

**Table 1 materials-16-04141-t001:** Concrete compressive strength experimental results.

Age of Concrete (Days)	Sample No.	Cube Compressive Strength $f_{ci,cub}$	Cube Mean Compressive Strength $f_{cm,cub}$	Concrete Density (kg/m^3^)
28	1	57.7	58.1	2320
	2	58.4		
45	1	60.1	61.4	
	2	62.7		
	3	61.4		

**Table 2 materials-16-04141-t002:** Prestressing steel experimental results.

Sample No.	Bar Section Area (mm^2^)	$f_{p0.1}$ (MPa)	$f_{p0.1m}$ (MPa)	$f_{pt}$ (MPa)	$f_{ptm}$ (MPa)	$E_{p}$ (GPa)	$E_{pm}$ (GPa)	$\varepsilon_{pt}$ (%)	$\varepsilon_{ptm}$ (%)
1	20.83	1802.00	1816.03	1944.07	1949.47	212.00	213.65	2.80	2.87
2	20.83	1807.70		1946.72		212.67		2.80	
3	20.83	1838.38		1957.62		216.28		3.00	

**Table 3 materials-16-04141-t003:** Tensile strength of reinforcement steel.

Type of Bar	Bar Section Diameter (mm)	Yield Limit $f_{y}\;\left(MPa\right)$	Tensile Strength $f_{t}\;\left(MPa\right)$
Longitudinal bars	10	540	615
Stirrups	6	532	620

**Table 4 materials-16-04141-t004:** Stress and strain values for prestressing steel at the characteristic points.

Characteristic Point No.	Stress (MPa)	Strain
1	1424.06	0.007
2	1816.03	0.010
3	1878.67	0.012
4	1949.47	0.0287

**Table 5 materials-16-04141-t005:** Strength and deformation parameters of steel components.

Type of Component	Modulus of Elasticity (MPa)	Yield Strength (MPa)	Elastic Strains	Ultimate Tensile Strength (MPa)	Ultimate Strains
Rebar H6	210,000	532 *	0.00253	620 *	0.08
Rebar H10	210,000	541 *	0.00258	615 *	0.08
Anchorage	200,000	430	0.00215	700	0.16

* Tested values according to Table 3.

**Table 6 materials-16-04141-t006:** Assumed CDP material model parameters.

Parameter	ψ (°)	ε	$\sigma_{b0}/\sigma_{c0}$	$K_{c}$	μ
Value	56.3 range: [10; 56.3]	0.1 range: [0.1; 1.0]	1.16	0.677 range: [0.5; 1.0]	0

**Table 7 materials-16-04141-t007:** Concrete model parameters.

Parameter	Value
$f_{c}=f_{cm}$	59.0 MPa
$f_{ct}=f_{ctm}$	4.095 MPa
$\;E_{c}=E_{cm,eff}$	21.05 GPa
$\beta_{ce}=\left[0.2;0.5\right]$	0.4
$\beta_{cr}=\left[0.05;0.4\right]$	0.1
$\beta_{ctr}=\left[0.05;0.2\right]$	0.1
$\varepsilon_{cf}$	3.45‰
$\varepsilon_{ctf}$	0.19‰
$\varepsilon_{cr}=\left[6.0;12.0\right]$‰	6.0
$\varepsilon_{ctr}=\left[0.3;0.6\right]$‰	0.42‰

## Data Availability

Not applicable.

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
