# Peer review of "Experimental and Numerical Research of Post-Tensioned Concrete Beams"

_materials, 2023, doi:10.3390/ma16114141_

Round 1
Reviewer 1 Report
Please check references citation in the text - lines 77, 94, 102, 120, 216, 478, 530.
Missing table references - lines 383, 416
Figure numbering and referencing - lines 327, 500, 587, 619, 645, 664, 721, 733, 738
Line 88-89 / Line 101 - what were the conclusions of the comparison between the numerical and experimental results?
Line 90 - what do you understand by a "node linear beam"?
Line 194 - please recheck the dimension with the data presented in Figure 1.
Figure 1 a and b - please recheck the dimensions on the left side of the beams: 200 mm = 95+60+60?
Line 213 - please remove the "shear" before the stirrups
Line 267 - there are no supports shown in Figure 5
Figure 5 - please indicate the loading points / transfer points from the steel beam to the concrete beam. What is the meaning of A0 and A letters? The grind seems non-symmetrical with respect to the mid span.
Lines 270-273 - what was the purpose of this paragraph? How is this relevant to the manuscript?
Figure 11 - what is the meaning of C1 and C2 other than the points corresponding to the peak load? Is there any other meaning of letter C?
Figures 12 and 13 are more or less similar. I would suggest keeping Figure 13 as the location of the strands is more clearly presented.
Line 374 - the authors stated that the yield and ultimate strength of longitudinal reinforcing bars and stirrups were experimentally determined (Table 3). Why would you consider the values given in the technical sheet?
Line 396 - the statement contradicts the caption of Table 6. Either the values were defined based on a certain scientific algorithm or they were assumed. Considering the subsequent information presented in the manuscript, I would suggest changing the caption of Table 4.
Line 428 - the concrete strength classes are the ones given in EC2. Maybe the authors tried to refer to a theoretical / possible strength class.
Lines 430, 544 - what is the meaning of the number following the equation number?
Please use the indicated type of citation in the manuscript and do not mix several types of citation. Try to rephrase the statements so that citations with [number] follow naturally.
Lines 75-76 - please rephrase, the sentence seems unfinished
Line 83 - please substitute "an" by "a"
Line 127 - please remove the second "also"
Line 153 - please check the spelling of "damage"
Line 246 - either the "strands were" or the "strand was"
Line 264 - pleas remove "an" if referring to both the support and the loading beams
Lines 763, 796 - what is the intended use of "exemplary"
Why "Authors" and not "authors"?
Reviewer 2 Report
An analytical review is not so much an analysis as a brief retelling of the achievements of other authors. It is desirable to show what has not been done by other authors in the direction of research. The experiment and the calculated experiment were performed correctly and have good convergence. The description of the results of the calculation experiment seems too detailed. It is necessary to check and correct the numbering of the figures.
Reviewer 3 Report
The behavior of post-tensioned concrete is complex and creating a representative FE model is challenging. The reviewer commends the authors’ on their experimental efforts and the FE modeling of this research.
Here are my comments:
1. It was not clear from the manuscript whether the single 0.6 in (15.2 mm) 7-wire strand was grouted or not? This needs to be clarified as the behavior of bonded grouted ducts is different from bonded non-grouted ducts.
2. The reviewer agrees that the non-prestressing bars and stirrups have an impact of the concrete behavior especially in the post-cracking phase and should be properly modeled.
3. Did the author use their model to predict the load deflection response of other tested beams from other researchers to verify the validity of their modeling? Doing that will make a stronger case for the assumptions used to model concrete behavior and interface bond before and after cracking.
4. How does the prestressing steel stress at failure (ultimate stress) from the tests and from the FE model compare to the equations proposed by Eurocode equations for ultimate design? I believe the FE model should be used to validate code design equations? Are these equations conservative? Not conservative? Can they be modified based on the FE work done in this research? Can the authors comment on that.
English languade is ok
Round 2
Reviewer 1 Report
The authors have addressed all issues raised during the reviewing process. I think the manuscript fulfills all the quality requirements to be published in its revised version.